# Development and validation of the leisure lifestyle and satisfaction assessment: A comprehensive tool for evaluating leisure engagement

En-Chi Chiu[1,2], Shu-Chun Lee[2,3]*

1 Department of Long-Term Care, National Taipei University of Nursing and Health Sciences, Taipei, Taiwan, 2 Department of Occupational Therapy, Taipei City Psychiatric Center, Taipei City Hospital, Taipei, Taiwan, 3 Department of Recreation and Sports Management, University of Taipei, Taipei, Taiwan

* A1057@tpech.gov.tw

## Abstract

Leisure activities play a pivotal role in enhancing the overall well-being and quality of life in people with schizophrenia. Leisure lifestyle and satisfaction provide important leisure-related information on how people with schizophrenia effectively engage in and benefit from leisure activities. The aim of the study was to develop a new measure with two sections (leisure lifestyle and leisure satisfaction): the leisure LIfestyle and SAtisfaction measure (LISA). Literature review, expert consultation, and cognitive interviews were conducted to develop and verify the content of the two sections. The leisure satisfaction section was examined for construct validity, Rasch reliability, and ceiling/floor effects. Eight experts reviewed the content, and 15 people with schizophrenia participated in a cognitive interview. Subsequently, 200 people with schizophrenia from one psychiatric center completed the two sections of the measure. The leisure lifestyle section comprised three items designed to assess personal values associated with engaging in leisure activities and preferences for leisure activities in the present and future. The leisure satisfaction section included 14 items to assess the level of satisfaction derived from engagement in leisure activities and demonstrated unidimensionality with infit and outfit mean squares ranging from 0.77 to 1.27 and 0.75 to 1.27, respectively, while the eigenvalue of the first contrast was 2.2. The leisure satisfaction section showed a sufficient Rasch reliability of 0.90 and no ceiling/floor effect (0.5–3.5%). The LISA can simultaneously assess leisure lifestyle and satisfaction, offering detailed insights into the leisure activities that people with schizophrenia are attracted to and their perceived enjoyment of these activities. The participants of this study were recruited from a single institution and we excluded people with schizophrenia with severe cognitive impairments, which may restrict generalizability. Future research is warranted to recruit people with schizophrenia from multiple institutions to cross-validate our findings.

**Data availability statement:** All relevant data are within the paper and its Supporting Information files.

**Funding:** Our study was funded by the National Science and Technology Council (NSTC), with grant numbers NSTC 114-2635-B-532-001 and MOST 108-2314-B-532-004.

**Competing interests:** The authors have declared that no competing interests exist.

## Introduction

Leisure is an activity pursued for its own enjoyment, undertaken during free time that is not allocated to mandatory responsibilities such as work, self-care, or sleep [1]. Schizophrenia is a serious mental disorder characterized by psychosis, emotional detachment, cognitive impairment, and social withdrawal [2]. Leisure activities play a crucial role in the well-being and quality of life of people with schizophrenia and have been employed as a method to remediate impairments [3,4]. Leisure pursuits provide opportunities for cognitive stimulation and emotional expression, which can mitigate cognitive and affective impairments in people with schizophrenia [5–7]. Participation in leisure activities can facilitate social interaction, which is essential for people with schizophrenia who experience social withdrawal and isolation [8,9]. The routine and structure of regular leisure activities contribute to stability and predictability, which are beneficial for individuals in managing the fluctuating nature of their symptoms [10,11]. Therefore, assessing leisure in people with schizophrenia is important for understanding their leisure interests and satisfaction, which can help clinicians and researchers set intervention goals and enhance participation in leisure activities.

Leisure is not only an activity, but also a structured and purposeful process that contributes to individuals' functional independence and quality of life within the framework of therapeutic recreation [8,12]. This perspective highlights the clinical significance of assessing leisure and aligns with the goals of therapeutic interventions for people with schizophrenia. Leisure lifestyle refers to a way of living in which leisure time is a significant part of daily routines [13]. Individuals prioritize activities that bring joy, relaxation, and personal fulfillment, which can balance work and responsibilities with pursuits such as hobbies, social interactions, and creative endeavors [14,15]. Embracing a proactive and enriched leisure lifestyle can foster well-being, reduce stress, and enhance overall happiness [16–18]. Leisure satisfaction describes the positive feelings and perceptions that individuals develop and experience by engaging in and choosing leisure activities [19]. A sense of satisfaction emerges when the leisure content and environment fulfill personal expectations [20,21]. Satisfaction is an important indicator of participation in leisure activities [22].

Some leisure measures are available to assess leisure-related information for people with schizophrenia, such as the Canadian Occupational Performance Measure (COPM) and the Leisure Satisfaction Scale (LSS) [23–25]. While the COPM and LSS assess leisure satisfaction, these two measures do not provide detailed information related to leisure lifestyle, such as regular participation in leisure activities, frequency of performing leisure activities, and leisure activities that individuals would like to engage in. Clinicians and researchers must gather comprehensive information on both leisure lifestyle and leisure satisfaction when designing interventions in clinical and research settings. However, current measures have not simultaneously captured these two critical sections (i.e., leisure lifestyle and leisure satisfaction) in a holistic manner, which results in an incomplete understanding of the full spectrum of leisure engagement. Therefore, the purpose of this study was to develop a new measure, the leisure LIfestyle and SAtisfaction measure (LISA), which can help understand

leisure lifestyle and leisure satisfaction in people with schizophrenia simultaneously, ultimately contributing to their overall well-being and quality of life.

## Materials and methods

### Participants

This study employed a methodological, cross-sectional design focusing on the development and validation of the LISA. People with schizophrenia from a psychiatric center in northern Taiwan were enrolled in the study. The criteria for inclusion were: (1) meeting the diagnostic criteria for schizophrenia as outlined in the Diagnostic and Statistical Manual of Mental Disorders, 5th edition; (2) age over 20 years; (3) maintaining a stable mental condition; and (4) the Mini-Mental Status Examination (MMSE) score ≥ 22. People with schizophrenia were excluded if they had a history of brain injury or were diagnosed with substance abuse or intellectual disability. A sample size of 200 was recommended when calibrating item parameters under the Rasch model [26,27]. This study was approved by the Taipei City Hospital Research Ethics Committee (TCHIRB-10412114). All participants provided written informed consent before participation.

### Procedures

The LISA was designed as a self-reported measure in three stages. The first stage involved a literature review that focused on leisure measures for people with disabilities. We designed the LISA with two sections: leisure lifestyle and leisure satisfaction. The leisure lifestyle section aimed to explore the personal value of engaging in leisure activities and preferences for leisure activities in both the present and future. The leisure satisfaction section aimed to assess the level of satisfaction derived from participating in the leisure activities. The second stage involved expert consultation. Eight experts, including occupational therapists, psychiatrists, and researchers specializing in psychometric properties provided comments on the suitability, wording, and relevance of the content, as well as scoring. After expert consultation, the leisure lifestyle section included three items: (1) level of importance of participating in leisure activities (5-point scale: 1 = extremely unimportant to 5 = extremely important); (2) regular participation in leisure activities and frequency of engagement; and (3) leisure activities respondents would like to engage in. The three leisure lifestyle items were designed as independent indicators to describe patterns of leisure engagement rather than to represent a single latent construct subjected to psychometric analysis. Thus, descriptive statistics were used to summarize responses. Leisure activities cover a wide range, and thus we classified the leisure activities of the third item in the leisure lifestyle section into eight categories. The eight categories were derived by synthesizing the classification reported in previous leisure studies [28,29], which groups activities based on their underlying purpose, such as entertainment, learning, physical health, creativity, intellectual stimulation, personal interest, and social interaction. These frameworks provide a theoretical basis for distinguishing leisure activities according to intention or motivation associated with participation. The eight categories were as follows: (1) audiovisual activities: watching movies, watching television, watching videos (e.g., on YouTube), surfing social networks (e.g., Facebook), playing video games, listening to music, and karaoke; (2) learning activities: reading, writing, language learning, photography, performing arts, tea ceremonies, book clubs, and attending lectures; (3) outdoor activities: walking, traveling, riding a bicycle, using roller skates, bird-watching, and camping, (4) intellectual pursuits: chess, playing cards, mahjong, sudoku, puzzles, magic, brain teasers, and Rubik's Cube; (5) artistic interests: playing musical instruments, art and cultural exhibitions, handicrafts, dancing, painting, and appreciating drama; (6) sports activities: ball sports, hiking, swimming, jogging, fitness, aerobic exercise, qigong, tai chi, Yuanji dance, and Neidan exercise; (7) hobbies: gardening, pet keeping, collecting items, and stamp collecting; (8) social activities: gathering with friends, family gatherings, hot spring bathing, shopping, religious activities and volunteer work (Appendix A). The leisure satisfaction category comprises 15 items. Items 1–14 used a 5-point scale: 1 = very dissatisfied, 2 = dissatisfied, 3 = neutral, 4 = satisfied, and 5 = very satisfied. Item 15 (I hope for improvements in my leisure life) was rated as either 0 = no or 1 = yes.

Cognitive interviews were conducted with 15 people with schizophrenia who met the inclusion and exclusion criteria. Cognitive interviews were conducted to gather feedback from people with schizophrenia on the clarity and comprehensibility of the content regarding leisure lifestyle and leisure satisfaction items. The third stage was the validation of the leisure satisfaction section of the LISA. Construct validity (e.g., unidimensionality), Rasch reliability, and ceiling/floor effects were evaluated. People with schizophrenia who met the inclusion and exclusion criteria were enrolled between October 2018 and April 2020.

## Measures

The MMSE assesses general cognitive functions [30]. This was used to verify whether the participants were able to follow the instructions provided. The total MMSE scores range from 0 to 30. Higher scores indicate better general cognitive function. The MMSE has shown adequate test-retest reliability in people with schizophrenia [31].

The Clinical Global Impressions-Severity scale (CGI-S) was used to evaluate the extent of psychiatric illness in participants with schizophrenia. The CGI-S is rated on a 7-point scale: 1 = no illness, 2 = borderline illness, 3 = mild illness, 4 = moderate illness, 5 = marked illness, 6 = severe illness, and 7 = extremely severe illness [32]. Substantial convergent validity of the CGI-S has been demonstrated in people with schizophrenia [33].

## Data analysis

Descriptive statistics (e.g., means and percentages) were used to illustrate the three leisure lifestyle items. The top ten most regularly participated leisure activities and the top ten leisure activities that participants wanted to engage in were represented.

Rasch analysis was conducted to assess the unidimensionality of the leisure satisfaction section. Infit and outfit statistics were used to evaluate whether the data fit a unidimensional Rasch model. Items with infit and outfit mean squares (MnSq) were less than 0.60 or higher than 1.40 demonstrating a misfit. Furthermore, principal component analysis of the standardized residuals was performed to verify unidimensionality. Unidimensionality was supported by an eigenvalue of the first contrast of < 3.0 [34]. Winsteps software was used to conduct Rasch analysis.

Once the unidimensionality of the leisure satisfaction section was established, Rasch reliability was calculated. The reliability criteria were ≥ 0.70 for comparisons among groups and ≥ 0.90 for comparisons among individuals [34]. Ceiling and floor effects were estimated using the percentages of participants who had the highest and lowest scores on the leisure satisfaction section, respectively. When the highest and lowest scores were ≥ 15% of all participants, this showed ceiling or floor effects, respectively [35].

## Results

In total, 200 people with schizophrenia completed the LISA, 44.5% of whom were male. The mean age of participants was 43.3 years. The average scores of the MMSE and CGI-S were 26.6 and 3.4, respectively. The demographic information of the participants is presented in Table 1.

For the three leisure lifestyle items, the mean score (standard deviation) for the importance of participating in leisure activities was 4.0 (1.0). Top ten of regularly participated leisure activities were watching television (18.0%), walking (8.8%), reading (7.8%), surfing social networks (7.3%), listening to music (5.9%), shopping (4.8%), ball sports (4.6%), watching videos (3.6%), playing video games (2.9%), jogging (2.9%), and religious activities (2.9%). The top ten leisure activities participants wanted to engage in were watching videos (3.9%), watching television (3.8%), family gatherings (3.7%), tea ceremonies (3.7%), shopping (3.2%), surfing social networks (3.1%), attending lectures (2.9%), hot spring bathing (2.8%), pet keeping (2.7%), and collecting items (2.4%).

Regarding the unidimensionality of the leisure satisfaction section, item 15 was deleted because the infit and outfit MnSqs were > 1.4. The infit and outfit MnSqs of item 1–14 were 0.77–1.27 and 0.75–1.27, respectively (Table 2). For

**Table 1. Participant characteristics (n = 200).**

| Characteristic | |
|---|---|
| Gender (%) | |
| Male | 89 (44.5) |
| Female | 111 (55.5) |
| Age (mean year [SD]) | 43.3 (11.1) |
| Onset age (mean year [SD]) | 21.4 (6.7) |
| Education (%) | |
| Elementary school | 6 (3.0) |
| Junior high school | 12 (6.0) |
| Senior high school | 96 (48.0) |
| College and above | 86 (43.0) |
| Type of antipsychotics (%) | |
| First generation | 39 (19.5) |
| Second generation | 101 (50.5) |
| Third generation | 21 (10.5) |
| Taking two types | 20 (10.0) |
| MMSE (mean point [SD]) | 26.6 (2.5) |
| CGI-S (mean point [SD]) | 3.4 (0.8) |
| Important on participating leisure activity (mean point [SD]) | 4.0 (1.0) |

SD: standard deviation; MMSE: Mini-Mental State Examination; CGI-S: Clinical Global Impressions-Severity scale.

**Table 2. Rasch statistics of leisure satisfaction section.**

| Item | Difficulty logit | Infit MnSq | Outfit Mnsq |
|---|---|---|---|
| 1. | 0.17 | 0.89 | 0.94 |
| 2. | −0.11 | 0.83 | 0.83 |
| 3. | 0.42 | 1.06 | 1.09 |
| 4. | 0.11 | 1.19 | 1.21 |
| 5. | 0.18 | 1.16 | 1.20 |
| 6. | 0.05 | 0.74 | 0.74 |
| 7. | 0.54 | 0.95 | 1.00 |
| 8. | −0.55 | 0.93 | 0.90 |
| 9. | −0.44 | 1.10 | 1.09 |
| 10. | 0.40 | 1.08 | 1.11 |
| 11. | −0.36 | 1.05 | 1.09 |
| 12. | −0.43 | 1.27 | 1.27 |
| 13. | 0.17 | 0.96 | 0.95 |
| 14. | −0.17 | 0.77 | 0.75 |

MnSq: mean square.

these 14 items, the eigenvalue of the first contrast was 2.2. The Rasch reliability was 0.90. No ceiling or floor effects were observed (0.5% and 3.5%, respectively). Appendix A presents the 14 items of the leisure satisfaction section.

## Discussion

The LISA contained two sections (leisure lifestyle and leisure satisfaction) that were designed to provide an overview of leisure interests and satisfaction with participation in leisure activities over the past month. The leisure lifestyle section sought to investigate the personal significance of engaging in leisure activities, and the leisure satisfaction section assessed the level of satisfaction derived from participating in leisure activities. Clinicians and researchers can use the LISA to obtain leisure-related information from people with schizophrenia.

In the leisure lifestyle section, participants perceived that engaging in leisure activities was important, as indicated by a mean score of 4 on a 1–5 scale, with higher scores reflecting greater perceived importance. Our findings showed that participants with schizophrenia regularly engaged in audiovisual activities (e.g., watching television, surfing social networks, listening to music, watching videos, and playing video games), which represented a higher proportion of their leisure activities. Besides audiovisual activities, they wanted to engage in social activities (e.g., family gatherings, shopping, and hot spring bathing), which offer opportunities for interpersonal interaction and socialization. Information on leisure interests can assist clinicians and researchers in designing leisure interventions aimed at enhancing the engagement and motivation of people with schizophrenia.

Unidimensionality implies that all items converge to assess a single construct [36]. The unidimensionality of the leisure satisfaction section was examined using Rasch analysis. The results provide evidence that the 14 items of the leisure satisfaction section assessed a single construct, and these items could be summarized as describing satisfaction with participating in leisure activities. The total score on the leisure satisfaction section ranged from 14 to 70. Higher scores indicated greater satisfaction.

Rasch reliability denotes the extent to which a measure is free from measurement error, reflecting the precision of the measurement [37]. A Rasch reliability value of 0.90 indicates excellent reliability, meaning that the leisure satisfaction section can reliably differentiate people with schizophrenia with different levels of leisure satisfaction. For clinical applications, a high reliability (≥ 0.90) supports the confident use of the scores for making individual-level decisions. Ceiling and floor effects can serve as evidence of a measure's effectiveness in distinguishing between the highest and lowest levels of a construct [38]. No ceiling or floor effects were observed in the leisure satisfaction section, demonstrating adequate discrimination ability in the highest and lowest scores of people with schizophrenia. Sufficient Rasch reliability and the absence of ceiling/floor effects underscore its precise and effective discriminatory ability in assessing satisfaction levels in people with schizophrenia.

The LISA is a self-reported measure used to comprehensively assess leisure in people with schizophrenia. It has three features that are applicable to people with schizophrenia. First, the LISA simultaneously assesses leisure interests and satisfaction. While existing measures emphasize leisure satisfaction, the LISA additionally assesses leisure participation values and preferences. The development of the LISA was supported by existing literature emphasizing the purposeful and health-promoting aspects of leisure. Second, it collects information on leisure activities in a structured form, which allows clinicians and researchers to obtain organized and clinically relevant data to guide intervention planning. Third, it is simple and easy to use in clinical setting, as the items are straightforward and completion time is short (10–15 minutes for the leisure lifestyle section and 5–10 minutes for the leisure satisfaction section). Therefore, the LISA can be considered as a practical and useful tool, as it enables efficient data collection without placing an excessive burden on the examinees and examiners.

This study has three notable limitations. First, we recruited participants from a single psychiatric center, which may have limited the generalizability of our findings. Future research should recruit people with schizophrenia from multiple psychiatric facilities to cross-validate these findings. Second, we excluded people with schizophrenia who had severe

cognitive impairments in addition, which may restrict the applicability of the results to broader schizophrenic populations. However, this exclusion was necessary because the LISA relies on self-reported leisure information and requires a certain level of cognitive function for accurate completion. Future research should consider the feasibility of using proxy reports for people with schizophrenia who have severe cognitive impairments. Third, this study did not examine the applicability of the LISA across different cultures and mental illness populations. Future research should explore the applicability of the LISA in diverse cultural contexts and investigate its psychometric properties in other mental illness populations.

## Conclusions

The LISA is a new measure designed to assess leisure lifestyle and satisfaction. The leisure lifestyle section provides descriptive information on individuals' perceived importance of leisure, frequency of participation, and preferred leisure activities, offering clinically relevant insights to guide intervention planning. The leisure satisfaction section has satisfactory construct validity, Rasch reliability, and no ceiling/floor effects. The LISA can provide valuable information for clinicians and researchers to tailor leisure interventions for people with schizophrenia. This study was conducted at a single institution and excluded people with schizophrenia with severe cognitive impairments, which may restrict the generalizability of our results. Future research is needed to validate the LISA across diverse clinical settings and populations to strengthen its applicability.

## Supporting information

**S1 File. Appendix A.**
(DOCX)

**S2 File. Score of subtests.**
(XLSX)

## Acknowledgments

We thank all the participants for their time and the research staff who helped collect the data.

## Author contributions

**Conceptualization:** En-Chi Chiu, Shu-Chun Lee.

**Data curation:** Shu-Chun Lee.

**Funding acquisition:** Shu-Chun Lee.

**Methodology:** En-Chi Chiu.

**Resources:** Shu-Chun Lee.

**Writing – original draft:** En-Chi Chiu.

**Writing – review & editing:** En-Chi Chiu, Shu-Chun Lee.

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
