## [Decision Letter · Decision Letter 0]

2 Jul 2025

Development and validation of the leisure lifestyle and satisfaction assessment: A comprehensive tool for evaluating leisure engagement

We look forward to receiving your revised manuscript.

Kind regards,

Yun-Ju Lai

Academic Editor

PLOS ONE

Journal Requirements:

2. Please describe in your methods section how capacity to provide consent was determined for the participants in this study. Please also state whether your ethics committee or IRB approved this consent procedure. If you did not assess capacity to consent please briefly outline why this was not necessary in this case.

4. Please include captions for your Supporting Information files at the end of your manuscript, and update any in-text citations to match accordingly. Please see our Supporting Information guidelines for more information: http://journals.plos.org/plosone/s/supporting-information .

Additional Editor Comments:

In addition to reviewers' comment, we still had some concerns. Given that this is a scale development study, the methodological framework appears insufficiently robust for such a purpose. While Rasch analysis is employed, the study would benefit from the inclusion of additional types of analyses to strengthen the validity and reliability of the scale. Relying solely on the accuracy of Rasch analysis results does not provide a comprehensive assessment of the instrument. Furthermore, the literature review section contains abrupt contextual shifts, which hinder the logical flow and coherence of the narrative. Transitions between sentences are often weak or underdeveloped, resulting in a fragmented reading experience. A more structured and cohesive presentation of the literature is recommended to enhance the clarity and academic quality of the manuscript.

Reviewers' comments:

Reviewer's Responses to Questions

**Comments to the Author**

1. Is the manuscript technically sound, and do the data support the conclusions?

Reviewer #1: Yes

Reviewer #2: Partly

2. Has the statistical analysis been performed appropriately and rigorously?

Reviewer #1: Yes

Reviewer #2: No

3. Have the authors made all data underlying the findings in their manuscript fully available?

Reviewer #1: Yes

Reviewer #2: Yes

4. Is the manuscript presented in an intelligible fashion and written in standard English?

Reviewer #1: Yes

Reviewer #2: Yes

Reviewer #1: I hope this letter finds you well. I had the opportunity to review your article titled, “Development and validation of the leisure lifestyle and satisfaction assessment: A comprehensive tool for evaluating leisure engagement”, which was submitted Plos One.

1. Abstract

- The abstract is structurally relatively consistent and includes elements of the research purpose, methods, results, and conclusion, so the flow of the writing is good.

- In particular, the research purpose (development of a new assessment tool, LISA) and the main assessment methods (including Rasch analysis) were clearly described.

- The development of a leisure participation measurement tool for patients with schizophrenia is a topic with practical and clinical usefulness and is considered to have high academic/clinical contribution.

- However, some sentences after the middle are redundant or have ambiguous word choices, making the meaning unclear.

- The fact that the study subjects were limited to patients from a single institution and the limitations in generalizability were not mentioned at all in the abstract.

2. Introduction

- The introduction clearly states the purpose of the study by emphasizing the importance of leisure activities in improving the quality of life of patients with schizophrenia.

- It is judged that the relevant literature persuasively expresses that leisure has a positive effect on cognitive, emotional, and social functions.

- It is considered to be academically outstanding that the concepts of leisure lifestyle and leisure satisfaction were independently defined and the influence of these two factors on mental health and quality of life was explained.

- It is considered desirable to suggest a starting point for research by mentioning existing tools such as COPM and LSS.

- At the end of the introduction, the research objective of ‘developing the LISA tool to evaluate the leisure lifestyle and satisfaction of patients with schizophrenia’ is clearly stated, which is expected to help readers understand.

- However, although sufficient evidence has been presented on the effectiveness of leisure activities, it is judged necessary to explain the specific limitations of existing leisure measurement tools (e.g. COPM, LSS, etc.).

- Lack of critical context as to why new tools are needed.

- We hope to emphasize a critical awareness of what specific academic or clinical gaps in the existing literature this study seeks to fill.

- We need to move beyond the simple ‘need’ level and make a compelling case for ‘why existing methodologies are not sufficient.’

3. Method

- The effort to secure the content validity and empirical legitimacy of the tool through a three-stage systematic tool development process (literature review → expert consultation → cognitive interview → main survey → validity/reliability verification) is considered appropriate and methodologically sound.

- The use of scale analysis based on the Rasch model has strengths in psychometric approaches, and the attempt to explain the internal consistency of the instrument through reliability and unidimensionality verification is impressive.

- The criteria for selecting participants (DSM-5-based diagnosis, MMSE ≥ 22, stable condition, etc.) are clear, and it is considered highly desirable to limit the subjects to those with secured cognitive ability.

- It is judged that the fact that the study was conducted separately as a cognitive interview (n=15) and a main survey (n=200) increased the design precision.

- Three leisure lifestyle items and 15 leisure satisfaction items are described separately, and the detailed item composition is also sufficiently explained.

- The results of statistical verification are presented in detail, including confirmation of unidimensionality through Rasch analysis (PCA eigenvalue = 2.2), appropriate infit/outfit range (0.75~1.27), reliability coefficient (Rasch reliability = 0.90), and absence of ceiling/floor effect, sufficiently supporting reliability and validity.

- However, the type of research design was not clearly specified.

- This study is classified as a measurement tool development study, but the absence of an explicit statement of the type of design may limit the reader's understanding.

- Since statistical basis for sample size (e.g. minimum required number of cases for Rasch analysis) is not provided, it is difficult for readers to independently judge the validity of the test.

- The theoretical basis or prior research on the criteria for classifying leisure activities is unclear.

- The leisure lifestyle (3 items) section relies only on technical summaries without statistical analysis, raising questions about the structural integrity of the entire tool.

4. Results

- Table 1 clearly presents basic demographic characteristics such as gender, age, education level, medication type, cognitive function (MMSE), and disease severity (CGI-S), thereby explaining how to judge the representativeness and heterogeneity of the sample.

- Based on the numerical values such as the average MMSE (26.6 points) and the average CGI-S (3.4 points), it can be seen that the overall cognitive level and symptom severity level of the subjects were good, so it was determined that the tool's self-reporting potential was secured.

- It is very impressive that the respondents emphasized practical significance by evaluating the importance of leisure activities (average 4.0), presenting the top 10 activities they actually performed, and the top 10 activities they hoped to participate in.

- The lifestyle items are judged to provide vivid data on user experience by combining qualitative and quantitative data.

- The results of Rasch analysis are statistically well presented.

- However, since only the frequency is presented for this item without statistical analysis, it is judged impossible to evaluate its structural validity as a measurement indicator.

- There is a lack of quantitative discussion on the interpretation of the results. For example, the meaning or clinical applicability of the Rasch reliability value of 0.90 is not specifically interpreted.

5. Discussion

- The discussion clarifies the purpose of the study by relatively clearly defining the connection between the research purpose (development of a leisure lifestyle and satisfaction measurement tool for patients with schizophrenia) and the main results (ensuring reliability and unidimensionality based on Rasch analysis, no ceiling/floor effect).

- It is judged that the strength of the logical development is that it emphasizes the two-sided evaluation function (interest activities vs. satisfaction) of the LISA tool and naturally highlights the possibility of actual clinical use.

- Some of the results, such as the proportion of audiovisual activities and the desire for social activities, implicitly suggest that they are similar to existing studies, thereby showing a certain degree of literary connection and thus making an academic contribution.

- The part in the latter half of the discussion that emphasized that LISA is simple and easy to use in clinical settings, taking only 10 to 15 minutes, can be said to be a good example of the possibility of practical use.

- Along with its advantages as a self-reporting tool, its ability to collect information on leisure activities of patients with schizophrenia in a structured form is considered a strength as a clinical tool.

- We clearly recognize the limitations of the sample size, which is limited to participants from a single institution, and the exclusion of patients with severe cognitive impairment. However, we highly value the suggestion of future multi-institutional studies and the possibility of introducing proxy reports.

- It is considered a desirable discussion to emphasize that while existing tools focused on leisure satisfaction, this tool has a practical contribution by measuring leisure participation values and preferences as well.

- It is judged that there is a lack of consideration of the cross-cultural applicability of LISA and its expandability, such as its applicability to various mental illness groups.

- It is judged that there is a lack of explanation as to what theoretical models (e.g., behavioral theory, self-determination theory, quality of life model, etc.) these contributions are based on and extended.

6. Conclusion

- It clearly summarizes the achievement of the core objective of this study (development of a LISA tool that can simultaneously evaluate the leisure lifestyle and satisfaction of patients with schizophrenia).

- In particular, it was mentioned that ‘construct validity, Rasch reliability, and no ceiling/floor effects’ were secured for the leisure satisfaction section, and the measurement completeness of the tool was concisely emphasized, which reaffirmed the necessity of this study.

- It is positive that the clinical practicality and interventional potential were briefly mentioned through the fact that the research results can be utilized by ‘clinicians and researchers to tailor leisure interventions.’

- However, the conclusion does not include any mention of the limitations of the study (single-center recruitment, exclusion of subjects with cognitive impairment, etc.), which may give the impression of over-exaggerating the scope and generalizability of the study.

- It is judged that there is a lack of a clear vision for the expandability and development direction of the research, as there are no suggestions for follow-up research (multi-institutional research, application to various groups, modification and supplementation of tools, etc.).

Reviewer #2: The topic addressed in the study is undoubtedly of interest and potential value to the field. However, the manuscript appears to lack sufficient methodological rigor, particularly in terms of statistical analysis. The procedures related to the development and validation of the measurement tool are either insufficiently described or altogether absent, which undermines the study’s credibility and replicability. In addition, the introduction section requires further elaboration. A more comprehensive and well-structured overview of the research background is necessary to clearly situate the study within the existing body of literature. In particular, the justification for the development of such a measurement instrument should be articulated in greater detail, emphasizing its theoretical and practical significance for the field.

**Do you want your identity to be public for this peer review?** For information about this choice, including consent withdrawal, please see our Privacy Policy

Reviewer #1: No

Reviewer #2: **Yes: ** Gul YAGAR

---

## [Author Response · Author response to Decision Letter 1]

13 Aug 2025

Reviewer #1: I hope this letter finds you well. I had the opportunity to review your article titled, “Development and validation of the leisure lifestyle and satisfaction assessment: A comprehensive tool for evaluating leisure engagement”, which was submitted Plos One.

1. Abstract

- The abstract is structurally relatively consistent and includes elements of the research purpose, methods, results, and conclusion, so the flow of the writing is good.

Reply:

Thank you for your valuable feedback.

- In particular, the research purpose (development of a new assessment tool, LISA) and the main assessment methods (including Rasch analysis) were clearly described.

Reply:

Thank you for your valuable feedback.

- The development of a leisure participation measurement tool for patients with schizophrenia is a topic with practical and clinical usefulness and is considered to have high academic/clinical contribution.

Reply:

Thank you for your valuable feedback.

- However, some sentences after the middle are redundant or have ambiguous word choices, making the meaning unclear.

Reply:

We have deleted the redundant words and revised the sentences in the abstract section as follows: “Subsequently, 200 people with schizophrenia from one psychiatric center completed the two sections of the measure.” (lines 29-31, page 2, abstract section)

- The fact that the study subjects were limited to patients from a single institution and the limitations in generalizability were not mentioned at all in the abstract.

Reply:

We have added this limitation in the abstract section as follows: “The participants of this study were recruited from a single institution and we excluded people with schizophrenia with severe cognitive impairments, which may restrict generalizability. Future research is warranted to recruit people with schizophrenia from multiple institutions to cross-validate our findings.” (lines 40-43, page 3, abstract section)

2. Introduction

- The introduction clearly states the purpose of the study by emphasizing the importance of leisure activities in improving the quality of life of patients with schizophrenia.

Reply:

Thank you for your valuable feedback.

- It is judged that the relevant literature persuasively expresses that leisure has a positive effect on cognitive, emotional, and social functions.

Reply:

Thank you for your valuable feedback.

- It is considered to be academically outstanding that the concepts of leisure lifestyle and leisure satisfaction were independently defined and the influence of these two factors on mental health and quality of life was explained.

Reply:

Thank you for your valuable feedback.

- It is considered desirable to suggest a starting point for research by mentioning existing tools such as COPM and LSS.

Reply:

Thank you for your valuable feedback.

- At the end of the introduction, the research objective of ‘developing the LISA tool to evaluate the leisure lifestyle and satisfaction of patients with schizophrenia’ is clearly stated, which is expected to help readers understand.

Reply:

Thank you for your valuable feedback.

- However, although sufficient evidence has been presented on the effectiveness of leisure activities, it is judged necessary to explain the specific limitations of existing leisure measurement tools (e.g. COPM, LSS, etc.).

Reply:

The COPM and LSS are used to assess leisure satisfaction. However, these two measures do not provide detailed information related to leisure lifestyle, such as regular participation in leisure activities, frequency of performing leisure activities, and leisure activities that individuals would like to engage in. We have revised related information in the introduction section as follows: “While the COPM and LSS assess leisure satisfaction, these two measures do not provide detailed information related to leisure lifestyle, such as regular participation in leisure activities, frequency of performing leisure activities, and leisure activities that individuals would like to engage in.” (lines 75-78, page 5, introduction section)

- Lack of critical context as to why new tools are needed.

Reply:

Clinicians and researchers must gather comprehensive information on both leisure lifestyle and leisure satisfaction when designing interventions. However, previous measures have not simultaneously captured these two critical sections (i.e., leisure lifestyle and leisure satisfaction) in a holistic manner. Therefore, the purpose of this study was to develop a new measure, the leisure LIfestyle and SAtisfaction measure (LISA), which can help understand leisure lifestyle and leisure satisfaction in people with schizophrenia simultaneously.

We have revised related information in the introduction section as follow: “Clinicians and researchers must gather comprehensive information on both leisure lifestyle and leisure satisfaction when designing interventions in clinical and research settings. However, current measures have not simultaneously captured these two critical sections (i.e., leisure lifestyle and leisure satisfaction) in a holistic manner, which results in an incomplete understanding of the full spectrum of leisure engagement. Therefore, the purpose of this study was to develop a new measure, the leisure LIfestyle and SAtisfaction measure (LISA), which can help understand leisure lifestyle and leisure satisfaction in people with schizophrenia simultaneously, ultimately contributing to their overall well-being and quality of life.” (lines 78-86, page 5-6, introduction section)

- We hope to emphasize a critical awareness of what specific academic or clinical gaps in the existing literature this study seeks to fill.

Reply:

Measures like the COPM and LSS only assess leisure satisfaction, which could not simultaneously capture information related to leisure lifestyle, such as regular participation in leisure activities, frequency of performing leisure activities, and leisure activities that individuals would like to engage in. This gap underscores the need for a more comprehensive measure, which can assess both leisure lifestyle and life satisfaction in a holistic manner. Clinicians and researchers must gather comprehensive information on both leisure lifestyle and leisure satisfaction when designing interventions in clinical and research settings.

We have revised related information in the introduction section as follows: “While the COPM and LSS assess leisure satisfaction, these two measures do not provide detailed information related to leisure lifestyle, such as regular participation in leisure activities, frequency of performing leisure activities, and leisure activities that individuals would like to engage in. Clinicians and researchers must gather comprehensive information on both leisure lifestyle and leisure satisfaction when designing interventions in clinical and research settings. However, current measures have not simultaneously captured these two critical sections (i.e., leisure lifestyle and leisure satisfaction) in a holistic manner, which results in an incomplete understanding of the full spectrum of leisure engagement.” (lines 75-83, page 5, introduction section)

- We need to move beyond the simple ‘need’ level and make a compelling case for ‘why existing methodologies are not sufficient.’

Reply:

While the COPM and LSS assess leisure satisfaction, they do not provide a comprehensive picture of leisure lifestyle, such as regular participation in leisure activities, frequency of performing leisure activities, and leisure activities that individuals would like to engage in. This lack of detail hampers the ability of clinicians and researchers to fully understand and address the diverse needs of people with schizophrenia. The existing measures are insufficient for capturing the full spectrum of leisure engagement.

We have revised related information in the introduction section as follows: “While the COPM and LSS assess leisure satisfaction, these two measures do not provide detailed information related to leisure lifestyle, such as regular participation in leisure activities, frequency of performing leisure activities, and leisure activities that individuals would like to engage in. Clinicians and researchers must gather comprehensive information on both leisure lifestyle and leisure satisfaction when designing interventions in clinical and research settings. However, current measures have not simultaneously captured these two critical sections (i.e., leisure lifestyle and leisure satisfaction) in a holistic manner, which results in an incomplete understanding of the full spectrum of leisure engagement.” (lines 75-83, page 5, introduction section)

3. Method

- The effort to secure the content validity and empirical legitimacy of the tool through a three-stage systematic tool development process (literature review → expert consultation → cognitive interview → main survey → validity/reliability verification) is considered appropriate and methodologically sound.

Reply:

Thank you for your valuable feedback.

- The use of scale analysis based on the Rasch model has strengths in psychometric approaches, and the attempt to explain the internal consistency of the instrument through reliability and unidimensionality verification is impressive.

Reply:

Thank you for your valuable feedback.

- The criteria for selecting participants (DSM-5-based diagnosis, MMSE ≥ 22, stable condition, etc.) are clear, and it is considered highly desirable to limit the subjects to those with secured cognitive ability.

Reply:

Thank you for your valuable feedback.

- It is judged that the fact that the study was conducted separately as a cognitive interview (n=15) and a main survey (n=200) increased the design precision.

Reply:

Thank you for your valuable feedback.

- Three leisure lifestyle items and 15 leisure satisfaction items are described separately, and the detailed item composition is also sufficiently explained.

Reply:

Thank you for your valuable feedback.

- The results of statistical verification are presented in detail, including confirmation of unidimensionality through Rasch analysis (PCA eigenvalue = 2.2), appropriate infit/outfit range (0.75~1.27), reliability coefficient (Rasch reliability = 0.90), and absence of ceiling/floor effect, sufficiently supporting reliability and validity.

Reply:

Thank you for your valuable feedback.

- However, the type of research design was not clearly specified.

Reply:

This study employed a methodological, cross-sectional design focusing on the development and validation of the LISA. We have added related information in the methods section as follows: “This study employed a methodological, cross-sectional design focusing on the development and validation of the LISA.” (lines 90-91, page 6, methods section)

- This study is classified as a measurement tool development study, but the absence of an explicit statement of the type of design may limit the reader's understanding.

Reply:

We have added the type of design in the methods section as follows: “This study employed a methodological, cross-sectional design focusing on the development and validation of the LISA.” (lines 90-91, page 6, methods section)

- Since statistical basis for sample size (e.g. minimum required number of cases for Rasch analysis) is not provided, it is difficult for readers to independently judge the validity of the test.

Reply:

A sample size of 200 was recommended when calibrating item parameters under the Rasch model1,2. We have added related information in the methods section as follows: “A sample size of 200 was recommended when calibrating item parameters under the Rasch model [25, 26].” (lines 97-98, page 6, methods section)

- The theoretical basis or prior research on the criteria for classifying leisure activities is unclear.

Reply:

The eight categories were derived by synthesizing the classification reported in previous leisure studies3,4, which groups activities based on their underlying purpose, such as entertainment, learning, physical health, creativity, intellectual stimulation, personal interest, and social interaction. These frameworks provide a theoretical basis for distinguishing leisure activities according to intention or motivation associated with participation.

We have added related information in the methods section as follows: “Leisure activities cover a wide range, and thus we classified the leisure activities of the third item in the leisure lifestyle section into eight categories. The eight categories were derived by synthesizing the classification reported in previous leisure studies [27, 28], which groups activities based on their underlying purpose, such as entertainment, learning, physical health, creativity, intellectual stimulation, personal interest, and social interaction. These frameworks provide a theoretical basis for distinguishing leisure activities according to intention or motivation associated with participation. The eight categories were as follows: (1) audiovisual activities: watching movies, watching television, watching videos (e.g., on YouTube), surfing social networks (e.g., Facebook), playing video games, listening to music, and karaoke; (2) learning activities: reading, writing, language learning, photography, performing arts, tea ceremonies, book clubs, and attending lectures; (3) outdoor activities: walking, traveling, riding a bicycle, using roller skates, bird-watching, and camping, (4) intellectual pursuits: chess, playing cards, mahjong, sudoku, puzzles, magic, brain teasers, and Rubik’s Cube; (5) artistic interests: playing musical instruments, art and cultural exhibitions, handicrafts, dancing, painting, and appreciating drama; (6) sports activities: ball sports, hiking, swimming, jogging, fitness, aerobic exercise, qigong, tai chi, Yuanji dance, and Neidan exercise; (7) hobbies: gardening, pet keeping, collecting items, and stamp collecting; (8) social activities: gathering with friends, family gatherings, hot spring bathing, shopping, religious activities and volunteer work (Appendix A).” (lines 118-137, page 7-8, methods section)

- The leisure lifestyle (3 items) section relies only on technical summaries without statistical analysis, raising questions about the structural integrity of the entire tool.

Reply:

The three items of the leisure lifestyle section were intentionally designed as descriptive indicators rather than as a unidimensional latent construct subjected to psychometric analysis. We have added related information in the methods section as follows: “After expert consultation, the leisure lifestyle section included three items: (1) level of importance of participating in leisure activities (5-point scale: 1 = extremely unimportant to 5 = extremely important); (2) regular participation in leisure activities and frequency of engagement; and (3) leisure activities respondents would like to engage in. The three leisure lifestyle items were designed as independent indicators to describe patterns of leisure engagement rather than to represent a single latent construct subjected to psychometric analysis. Thus, descriptive statistics were used to summarize responses.” (lines 112-118, page 7, methods section)

4. Results

- Table 1 clearly presents basic demographic characteristics such as gender, age, education level, medication type, cognitive function (MMSE), and disease severity (CGI-S), thereby explaining how to judge the representativeness and heterogeneity of the sample.

Reply:

Thank you for your valuable feedback.

- Based on the numerical values such as the average MMSE (26.6 points) and the average CGI-S (3.4 points), it can be seen that the overall cognitive level and symptom severity level of the subjects were good, so it was determined that the tool's self-reporting potential was secured.

Reply:

Thank you for your valuable feedback.

- It is very impressive that the respondents emphasized practical significance by evaluating the importance of leisure activities (average 4.0), presenting the top 10 activities they actually performed, and the top 10 activities they hoped to participate in.

Reply:

Thank you for your valuable feedback.

- The lifestyle items are judged to provide vivid data on user experience by combi

---

## [Decision Letter · Decision Letter 1]

11 Sep 2025

Development and validation of the leisure lifestyle and satisfaction assessment: A comprehensive tool for evaluating leisure engagement

Dear Dr. Lee,

Thank you for submitting your manuscript to PLOS ONE. After careful consideration, we feel that it has merit but does not fully meet PLOS ONE’s publication criteria as it currently stands. Therefore, we invite you to submit a revised version of the manuscript that addresses the points raised during the review process.

We look forward to receiving your revised manuscript.

Kind regards,

Yun-Ju Lai

Academic Editor

PLOS ONE

Journal Requirements:

Reviewers' comments:

Reviewer's Responses to Questions

**Comments to the Author**

Reviewer #2: (No Response)

2. Is the manuscript technically sound, and do the data support the conclusions?

Reviewer #2: Partly

3. Has the statistical analysis been performed appropriately and rigorously?

Reviewer #2: Yes

4. Have the authors made all data underlying the findings in their manuscript fully available?

Reviewer #2: Yes

5. Is the manuscript presented in an intelligible fashion and written in standard English?

Reviewer #2: Yes

Reviewer #2: Review Report

I would like to thank the authors for providing a comprehensive response to the study and for making the necessary revisions in line with the reviewers’ comments and suggestions. The additional sections have contributed to strengthening the overall integrity of the research.

I believe that supporting this valuable study, which is intended to be introduced to the literature, with a stronger theoretical foundation will enhance its academic rigor. The main focus of the study, which examines individuals with schizophrenia within the context of therapeutic recreation as a functional aspect of leisure, is of significant importance. Therefore, it is recommended that, in the introduction, the concept of leisure be discussed not in its general sense but specifically within the framework of therapeutic recreation. Such an approach would improve both the coherence of the work and its original contribution. The literature review presented in the Procedures section may serve as an important reference point for the authors in this regard.

The main shortcoming of the study appears to stem from the excessive generalization of the topic. For this reason, it is suggested that the structure of the study be reorganized within a stronger theoretical framework. The recommended revisions are not related to analyses or procedures but rather to the need for a more robust theoretical grounding and for addressing leisure from a functional perspective.

When evaluated in terms of limitations, I believe that by removing the fourth limitation, the scope of the study could be expanded. This would also provide a valuable guide for future research.

Overall, the study has the potential to make a meaningful contribution to the literature. However, the suggested revisions would substantially strengthen its academic depth and enhance its value to the field.

**Do you want your identity to be public for this peer review?** For information about this choice, including consent withdrawal, please see our Privacy Policy

Reviewer #2: **Yes: ** Gül Yağar

---

## [Author Response · Author response to Decision Letter 2]

26 Oct 2025

Reviewer #2: Review Report

I would like to thank the authors for providing a comprehensive response to the study and for making the necessary revisions in line with the reviewers’ comments and suggestions. The additional sections have contributed to strengthening the overall integrity of the research.

I believe that supporting this valuable study, which is intended to be introduced to the literature, with a stronger theoretical foundation will enhance its academic rigor. The main focus of the study, which examines individuals with schizophrenia within the context of therapeutic recreation as a functional aspect of leisure, is of significant importance. Therefore, it is recommended that, in the introduction, the concept of leisure be discussed not in its general sense but specifically within the framework of therapeutic recreation. Such an approach would improve both the coherence of the work and its original contribution. The literature review presented in the Procedures section may serve as an important reference point for the authors in this regard.

Reply:

We appreciate your recommendation to frame the concept of leisure more specifically within the context of therapeutic recreation, rather than in a general sense. We have added related information in the introduction section as follows: “Leisure is not only an activity, but also a structured and purposeful process that contributes to individuals’ functional independence and quality of life within the framework of therapeutic recreation [8, 12]. This perspective highlights the clinical significance of assessing leisure and aligns with the goals of therapeutic interventions for people with schizophrenia.” (lines 64-68, pages 4-5, introduction section)

The main shortcoming of the study appears to stem from the excessive generalization of the topic. For this reason, it is suggested that the structure of the study be reorganized within a stronger theoretical framework. The recommended revisions are not related to analyses or procedures but rather to the need for a more robust theoretical grounding and for addressing leisure from a functional perspective.

Reply:

Thank you for this insightful comment. While the LISA was not developed based on a single formal theoretical model, its development was supported by existing literature emphasizing the purposeful and health-promoting aspects of leisure participation. We have revised related information in the discussion section as follows: “The LISA is a self-reported measure used to comprehensively assess leisure in people with schizophrenia. It has three features that are applicable to people with schizophrenia. First, the LISA simultaneously assesses leisure interests and satisfaction. While existing measures emphasize leisure satisfaction, the LISA additionally assesses leisure participation values and preferences. The development of the LISA was supported by existing literature emphasizing the purposeful and health-promoting aspects of leisure.” (lines 237-242, pages 13-14, discussion section)

When evaluated in terms of limitations, I believe that by removing the fourth limitation, the scope of the study could be expanded. This would also provide a valuable guide for future research.

Reply:

We have removed the fourth limitation.

Overall, the study has the potential to make a meaningful contribution to the literature. However, the suggested revisions would substantially strengthen its academic depth and enhance its value to the field.

Reply:

We sincerely appreciate the reviewer’s thoughtful and encouraging comments regarding the potential contribution of our study.

---

## [Decision Letter · Decision Letter 2]

18 Nov 2025

Development and validation of the leisure lifestyle and satisfaction assessment: A comprehensive tool for evaluating leisure engagement

PONE-D-25-19851R2

Dear Dr. Lee,

We’re pleased to inform you that your manuscript has been judged scientifically suitable for publication and will be formally accepted for publication once it meets all outstanding technical requirements.

Kind regards,

Yun-Ju Lai

Academic Editor

PLOS ONE

Reviewers' comments:

Reviewer's Responses to Questions

**Comments to the Author**

Reviewer #1: (No Response)

Reviewer #2: All comments have been addressed

2. Is the manuscript technically sound, and do the data support the conclusions?

Reviewer #1: (No Response)

Reviewer #2: Yes

3. Has the statistical analysis been performed appropriately and rigorously?

Reviewer #1: (No Response)

Reviewer #2: Yes

4. Have the authors made all data underlying the findings in their manuscript fully available?

Reviewer #1: (No Response)

Reviewer #2: Yes

5. Is the manuscript presented in an intelligible fashion and written in standard English?

Reviewer #1: (No Response)

Reviewer #2: Yes

Reviewer #1: - The revised introduction defines leisure as “functional independence and quality of life within the framework of therapeutic recreation,” addressing the theoretical weaknesses pointed out by previous reviewers.

- This revised study is considered excellent in that it goes beyond simply describing the importance of leisure participation and re-examines the functional meaning of leisure in a rehabilitation and therapeutic context.

- In addition, the research purpose of distinguishing between leisure lifestyle and leisure satisfaction and measuring them in an integrated manner was presented more persuasively.

- The revised manuscript clearly describes the ‘3-step procedure (literature review – expert review – cognitive interview – Rasch analysis)’ and strengthens the logical connection between each step.

- In particular, the division of leisure activities into eight categories (audiovisual, learning, outdoor, intellectual, artistic, sports, hobbies, and social activities) can be seen as a structural approach that ensures ‘content validity.’

- Verification of unidimensionality, reliability (0.90), and ceiling/floor effects (absence) through Rasch analysis proved the scientific precision of the measurement tool.

- In the revised discussion section, the three core features of LISA (① simultaneous assessment of leisure interest and satisfaction, ② provision of structural and clinical information, ③ simplicity) were clearly organized, and the applicability in clinical settings (completion within 10-15 minutes, minimizing patient burden) was emphasized.

- It is judged that these contents clearly presented the academic contribution of the paper as well as its clinical usefulness by highlighting its practical value and usability as a rehabilitation treatment tool.

- It is impressive that the unnecessary fourth limitation was deleted as advised by the reviewer, and the remaining three limitations were specified.

- In particular, the direction of follow-up research was clearly presented by specifying the need for multi-institutional samples and cross-cultural validation.

- It is judged that the mention of the possibility of ‘proxy report’ while recognizing the limitations of self-report shows practical insight.

Reviewer #2: I thank the authors for their detailed discussion of the evaluations. Their responses are sufficient. While links to sources used in the bibliography appear in the citation section on their pages, I believe they could also be included here (e.g., reference 26).

**Do you want your identity to be public for this peer review?** For information about this choice, including consent withdrawal, please see our Privacy Policy

Reviewer #1: No

Reviewer #2: **Yes: ** Gül Yağar

---

## [Editor Report · Acceptance letter]

PONE-D-25-19851R2

PLOS One

Dear Dr. Lee,

I'm pleased to inform you that your manuscript has been deemed suitable for publication in PLOS One. Congratulations! Your manuscript is now being handed over to our production team.

Kind regards,

on behalf of

Dr. Yun-Ju Lai

Academic Editor

PLOS One